# Creative Intercorporeality in Collaborative Work of Choreographers with and without Disabilities: A Grounded Theory Approach

**DOI:** 10.3390/ijerph19095548

**Published:** 2022-05-03

**Authors:** Jian Kim, Jooyeon Jin, Aeryung Hong

**Affiliations:** 1Department of Sports and Dance, Sangmyung University, Seoul 03016, Korea; artsedu@smu.ac.kr; 2Department of Sport Science, University of Seoul, Seoul 02504, Korea; jjin13@uos.ac.kr; 3Global Research Institute for Arts and Culture Education, Sangmyung University, Seoul 03016, Korea

**Keywords:** creative intercorporeality, collaborative work, choreographers with and without disabilities, social cognitive theory

## Abstract

The purpose of this study was to present an academic discourse on a theoretical framework and acceptance process of ‘creative intercorporeality’ in the collaborative work of choreographers with and without disabilities. To this end, a grounded theory approach using a qualitative research method was employed to dancers who have participated in collaborative choreography. This study employed the perspective of social cognitive theory about the process in which dancers with and without disabilities form emotional empathy and trust relationships through continuous interactions for creative work. Physical, emotional, cognitive, and behavioral empathy and interactions in the collaborative work of choreographers with and without disabilities were discussed as a process of forming ‘creative intercorporeality’ that is defined as creative attitude and perspective consisting of harmony, concurrency, consistency, and balance.

## 1. Introduction

Choreography is an art that designs a series of human movements and the form is defined within movements [1,2]. Choreography is not just activity using body, but activity created through intellectual processes to express one’s emotions and intentions using bodily movements [3]. It spans various fields of arts and sports such as dance, theater, musical, opera, gymnastics, figure skating, cheerleading, and artistic swimming. In particular, choreography created by two or more choreographers must contain dynamic exchange in intellectual capability, interact through bodily movements, and form creative solidarity. In this regard, well-developed collaborative choreography may include in-depth work and processes that harmonize methods of movement and expression by interacting cognitive, relational, aesthetic, and creative thinking skills among choreographers.

In the last two decades, collaborate work of choreographers with and without disabilities were actively attempted in different settings worldwide [4]. In addition, there have been recent scholarly activities dealing with topics such as experiences of children with disabilities in elementary school dance education [5,6,7,8], collaborative dance lessons in inclusive education environment [9], cooperative programs in traditional dance [10,11] and creative practice of dance artists [12,13]. These endeavors may guide, from the perspective of individuals with disabilities, how to overcome a lack of confidence and empowerment caused by disabilities, and from the perspective of individuals without disabilities, how to understand disabilities and develop an attitude away from prejudice. For these reasons, collaborative work of choreographers with and without disabilities is probably promising as it results in forming a positive attitude that recognizes and embraces the differences.

Meanwhile, most participation of people with disabilities in dance has been regarded as part of amateur or community dance rather than the area of “professional”. The potential reason, regardless of the culture of the East and the West, might be that the audience has been only used to watching the dance performed by dancers with the best physical conditions at the best theater. In other words, prejudice that limits the artistic value of dance, centering on physical figures and dance movements of dancers without disabilities, has been socially prevalent for quite a long time. However, breaking away from this prejudice, over the past decade, the development of “adapted dance” and/or “physically integrated dance” has gradually been expanded from an inclusive perspective [4,14]. This is probably influenced by a disability paradigm shift to perceive the concept of disability as a social and non-categorical phenomenon rather than a medical and categorical phenomenon. This change may prompt an contextual extension of dance participation towards inclusive environment where dancers with and without disabilities perform together [15,16,17,18].

The goal of the inclusive perspective in dance is to break up the technical framework of people without disabilities in locomotion and balance, and to expand the language of dance movements in various ways, so that people with disabilities can enter the norm of choreography more actively [19]. For example, dancesport has been widely known as a representative genre of dance that shows inclusive and cooperative characteristics as at least one of the dancers is in a wheelchair. Even the team of dancers with and without disabilities perform all modern genres such as Tango, Waltz, and Quickstep, and Latin American dances such as Chacha, Samba, Pasodoble, Lumba, and Jive are called “Combi-dance”. This collaborative choreography is not limited to the genre of dance in which the step routine is determined by certain criteria. The meaning of “creative integration” based on atypicality and possibility is important in that various creative elements must be drawn and new compositions must be created cooperatively. In the inclusive dance, directors and choreographers tend to focus on what dancers with disabilities can do in new and alternative ways, rather than focusing on what they cannot do. Due to the nature of inclusive environment, choreographers with disabilities may develop dance repertoire regardless of what disability conditions and degrees they have and collaborate through improvisation, and make new movements with their peers without disabilities.

However, there may be a dilemma in expressing “integrated” or “inclusive” dances through the performance of dancers with disabilities on a stage [4,20,21]. For example, it is necessary to take into account shortcomings such as blind access to infinite possibilities [22], directing dancers with disabilities treated as stage props, injuries and extreme damage caused by excessive technical attempts, and targeting audiences with disabilities rather than without disabilities. Nevertheless, it should be noted that the cooperative choreography is still as important as the serious creative process itself [23]. In inclusive dance process, one may have the following questions: “What movements can dancers with disabilities make?”, “How do you communicate among dancers with and without disabilities,” and “How do you solve the problem of movement for possible dance?” To answer those questions, cooperative choreography among dancers with and without disabilities aims for a more professional performance and presents a new vision for a “barrier-free society” [24].

The collaborative choreography goes through a cognitive process [25] that strives to empathize with each other and form a relationship for physical and emotional interactions; that is, bonding, intimacy, and trust, which soon shifts to movements as a dance language [26,27,28,29]. This process is explained by the concept of interaction that emphasizes the essential role of individuals with different abilities in the process of social understanding [30]. This notion is supported from the perspective of social cognitive theory, assuming that human behavior is a function of interactions of cognitive process and environment in a heterogeneous way, and the quality of human behavior is determined by experience in the particular environment [31,32,33,34,35,36,37].

With this core concept of the theory, Shogo Tanaka attempted to reexamine the concept of Merleau-Ponty’s intercorporeality and specify it as a extended social cognitive theory [38,39]. He developed the concept of behavioral unity, primitive empathy, interactive synchronization, and a sense of mutual understanding. Based on the perspective of this cognitive science, intercorporeality is defined as a shared perception including emotions and thoughts of two independent individuals. This can be said to be the origin of empathy, arising from the connection of cognition and behavior among oneself and another self; that is, the interrelationship among one’s body and another’s body.

The purpose of this study was to clarify the meaning of the cooperative choreography process and experience of choreographers with and without disabilities in the context of “creative intercorporeality” and to re-examine the discourses related to the construct. To achieve the purpose, end, this study employed a grounded theory approach as a qualitative research method to build empirical knowledge and a theoretical framework for dancers who have participated in integrated dance, especially collaborative choreography.

## 2. Research Method

This study employs a grounded theory approach to examine the process by which choreographers with and without disabilities embrace creative intercorporeality for collaborative work. Based on the experiences of the choreographers, this study examined how adapted dance affects disability awareness in the context of creative intercorporeality, and how a choreographer’s work embrace and transform a disability in the context of creative intercorporeality. The Grounded Theory, which was developed by sociologists Glaser and Strauss, was used for developing a theoretical model that explains interactions concerning systematic and comprehensive processes [40].

### 2.1. Research Participants

Ten participants who have experiences working with performers and/or choreographers with and without disabilities were conveniently recruited from an annual dance festival funded by Korean government. Demographic information of the participants are shown in Table 1. The approval (SMUIRB C-2020-017) from the Institutional Review Board (IRB) was obtained and the consent was collected from all research participants before data collection was initiated to ensure research ethics. 

### 2.2. Data Collection and Analysis

Data were collected from November 2020 to January 2021 through in-depth interviews using a semi-structured questionnaire. For a research participant who is overseas and not available for in-depth interviews in person, the researchers collected data via emails. The semi-structured questionnaire covered four areas: introduction to disability art and the artist’s portfolio, experience participating in disability art, difficulties in the collaborative work, and social recognition of disability and creative inter-corporeality. In order to collect data, an average of 1.5 in-depth interviews were conducted with the study participants for 1 hour 30 minutes to 2 hours, and the consent of the study participants was obtained and recorded. All interview contents were transcribed and coded and classified into upper and lower categories. 

Data analysis was conducted according to the analytical methods of open coding, axial coding, and selective coding presented by Strauss and Corbin [41]. Through open coding, 190 significant concepts were classified among the interview contents, and 30 subcategories and 14 categories were categorized centering on similar concepts. Conditions, phenomenon, strategy, and sequences constituting the paradigm model were classified as axial coding. In the selective coding process, the core category and the entire story line are summarized. After analysis, the details of descriptions were confirmed by the researchers, and the names of the participants were changed to pseudonyms or initials.

## 3. Result (1) Concepts and Categories

To examine the process by which choreographers who have experiences working with ones with and without disabilities accept creative intercorporeality, 190 conceptual words were derived as a result of analyzing the data line by line. Those with similar meanings were gathered to derive 30 subcategories, from which those with similar meanings were placed into superordinate categories, deriving a total of 14 categories.

Among the experiences of choreographers with and without disabilities as a process of acceptance of creative intercorporeality was the “creative intercorporeality through integrated dance,” which is a central phenomenon, as an interaction between causal conditions and contextual conditions; to overcome its difficulties, choreographers used action/interaction strategies. The process of reaching consequences as the impact of intervening conditions was determined. As a result of the analysis, the paradigm model on the collaborative choreography process was developed to demonstrate creative intercorporeality with subcategories. 

### 3.1. Causal Conditions

Causal conditions indicate the conditions or events that lead a certain phenomenon to occur, lay the groundwork, or develop. The acceptance process of the collaborative work of choreographers with and without disabilities begins from new motivation in choreography; more specifically, choreographers pondered over the essence of dance and continuity in exploring the body and implemented work extended by a new perspective on dance. They attempted professional–nonprofessional movements in dance styles based on movements as the collaborative choreography began to contemplate the disability discourse in dance. The choreographers served as a link between works with artistic value and the public sector through adapted dance, conducting various activities in events hosted by the government and local governments. These activities are exposed to society due to these internal and external public events, and further opportunities bring more positive effects. As such, the awareness and consideration of the context of creative intercorporeality began at the same time that choreography work was born as an adapted dance. Therefore, elements of causal conditions could be categorized into personal situations such as ‘new approach to the body’ and ‘motivation for new ideas’.


*We spent a few weeks getting to know each other, introducing and talking about performances of a dancer with a disability, watching performance videos, and moving our bodies together. As we experienced trial and error in experiencing new things, we started to change little by little. Remembering the elements of impromptu amusement that I learned from the Axis Dance Company, I applied the improvisation method to our class and witnessed the stages of transformation in our art.*
(Research participant B)

### 3.2. Contextual Conditions

Contextual conditions refer to the structural context that pertains to a certain phenomenon and the specific conditions for action and interaction. As a result of analysis, “mirroring others,” “breaking the pattern of the body,” and “shifting to a paradoxical way of thinking” were required as a “methodological agenda.” It was necessary to break the stereotypes between choreographers for the collaborative work, and to communicate by determining the movements within the scope of potential capability. As such, the methodological agenda about diversity and integration accepts disabilities in the context of creative intercorporeality and provides the conditions of action and interaction regarding the process of integration required. Therefore, elements of context conditions could be categorized into personal situations such as ‘acquiring mutually integrated choreography’ and ‘methodological agenda of support system’.


*By showing the public the dance that engages our body most actively, it became a determining factor in breaking down the barrier between disability and art.*
(Research participant F)

### 3.3. Central Phenomenon

Central phenomenon is defined as how a certain phenomenon is handled or progressed, and experience or core issue commonly shared by the research participants, as well as a key event to which action and interaction are related. As a result of the analysis, awareness of creative intercorporeality in choreography was the core phenomenon and was more specifically classified into balanced exploration of disability related expressions, empathy from potential behavior, and acquiring mutually integrated choreography. This experience was became an important turning point for the research participants because they were able to develop a new perspective on disabilities, breaking their prejudice or bias after participation, claiming that they attempted to find the balance point in expression or lead the space through potential behavior. In particular, “creative intercorporeality through integrated dance” was the consequence of causal conditions and described by the participants as an important disability art experience in the context of creative intercorporeality; thus, it was presented as the central phenomenon.


*With poliomyelitis, my body is in discomfort, and the center of gravity is different for me. In other words, my range of motion relying on a shifted center of gravity and the texture of my motion from different postures are completely different compared to others. However, I just take it as differences amongst performers and I think that can be the point of attraction. Would you say there is any difference between a student with a curved pelvis lifting a leg versus a person with a completely distorted center of gravity lifting a leg? When it comes to the performance of lifting a leg, there must be wholesome respect for the body.*
(Research participant C)

### 3.4. Intervening Conditions

Intervening conditions are the structural and extensive conditions affecting action and interaction, as well as the phenomenon. The research participants responded that inadequate support system and prejudice against adapted dance created barriers and frustrations against choreography. They also highlighted difficulties in the collaboration process as clash of opinions and complaints, which were occasionally raised between choreographers with and without disabilities in the choreography process. Moreover, when contacting with partners and selecting a partner in group dance moves they experienced difficulties due to physical and mental resistance; the safety of dancers also had to be considered. Therefore, elements of intervening conditions could be categorized into personal situations such as ‘barrier and prejudice in choreography’, ‘conflicts in collaboration’, and ‘risk of injury and resistance’.


*While working, disability was never a reference point for me. A person with a disability is the same human being as us, and I found it ironic labelling people with disabilities based on the perception that people without disabilities are the norm and the rest are not. It’s merely a little difference, but we all are the same in talking to each other, sharing our minds, and expressing ourselves using our bodies as the medium.*
(Research participant B)

### 3.5. Action/Interaction Strategies

Action/interaction strategies are a method to handle a problematic phenomenon, and they refer to intentional and deliberate behaviors that must be taken to solve a problem in practice. As for the new approach to the body, the research participants reset their perspectives and premises regarding how impairments in body functions and structures are perceived by society and other people, and whether they could head in a positive direction. They also had to communicate endlessly and show respect among dancers and made constant attempts to see how movements were interpreted from various perspectives. “Mutual understanding and respect” was a strategy for work by the participants within adapted art to proceed in a more meaningful and appropriate direction in the context of creative intercorporeality. Based on communication and respect, as well as empathy and trust to understand differences, the research participants pursued ways to head in a new direction to accept the differences, breaking away from the negative perspective on these in bodies and movements. Therefore, choreographers working their peers with and without disabilities were accepting it by using strategies such as the “exploration of a new process of approach” and “mutual understanding and respect”.


*I believe that artists and dancers with disabilities daily experience unique challenges that derive from their discomfort. This is why I see disability art as a form of art that acknowledges the differences in each person. There will be uniqueness from such discomfort and differences, and the empathy that deems such uniqueness as the novelty will arise.*
(Research participant C)

### 3.6. Consequences

Consequences indicate outcomes or results of action/interaction to cope with a certain phenomenon or to manage and maintain such the phenomenon. The experience of accepting creative intercorporeality for choreographers with and without disabilities participating in the collaborative work resulted in the consequences of “synchronizing and empathy”, “creative reciprocity and interaction”, “discovery/re-discovery of potential”, and “future sustainability of disability art”.

The strategy “creative reciprocity” explored coexistence through creative expression and discovered what creates the conditions for inclusive dance by acquiring trust and empathy from reciprocity. “Interactive synchronization and empathy” showed how the research participants discovered new interpretations of the body and various possibilities of movements through disability art, instead of considering a disability equals to incompetence. Moreover, “future sustainability of disability art” indicates that, for continuous disability art activities, it is necessary to activate actual projects in addition to policy support, and to establish an inclusive environment for the coexistence of dancers.


*Even in adapted dance, I am not moved by seeing dancers with disabilities on the stage, but purely by the work itself and the competence of those dancers. I want to say that, to change the stereotypical perception, dancers with disabilities need to grow their competency in creating a balance and harmony with choreographers and performers without disabilities.*
(Research participant F)

## 4. Result (2) Core Category and Situation Model

### 4.1. Core Category

This study conducted selective coding for the acceptance process of the research participants in collaborative work of choreographers with and without disabilities in the context of creative intercorporeality, ultimately forming a paradigm model and the story outline. As a result, the core category was “creative intercorporeality in dancers with and without disabilities”.

Through such a process, it was possible to determine how the integrated approach that embraces is due to the collaborative work of choreographers with and without disabilities, and how the collaboration is formed. 

First, choreographers began to look for new works originating from fundamental concerns such as exploration of the body, audience development, and dance. While the focus had been on working with dance specialists before, now the targets changed to various non-specialists such as an old lady, a middle-aged man, or a teenager, thereby expanding the scope of activities and naturally participating in collaborative work of choreographers with and without disabilities. Some dancers participated with interest in adapted dance after watching a performance by overseas a dancer with a disability, and others were recommended by other dancers who had already participated in adapted dance or began disability art upon the request of an adapted dance organization. Participating in collaborative work of choreographers with and without disabilities could be regarded as a starting point for many choreographers and performers to start a new form of work through various types of approaches, while also having the opportunity to gain disability awareness.

Second, the choreographers considered that disability awareness is important for participating in disability art. However, since education on disability awareness has not existed for a long time, many individuals without a disability still cannot easily approach to a disability condition. Nevertheless, the research participants showed an immediate response to change through education on disability awareness. The research participants who have collaborative work experience of choreographers with and without disabilities were experiencing change in naturally accepting disabilities in education on disability awareness.

Third, the research participants were able to take part in national events and projects of public institutions as an opportunity to expose a disability using disability art as a medium. With national and public art events, it was possible to not only be exposed to many people but also approach them in a less awkward way. Currently, there is a great interest in cultural diversity worldwide, and South Korea is also making a move to extensively improve disability awareness through policies and projects of public institutions. In this process, dancers linked their work with artistic value to the public sector, thereby creating an opportunity to promote positive disability awareness through internal and external public events.

Fourth, for the research participants who have experience in collaborative work, the form of disability and ability was an opportunity to have a new understanding of what was becoming meaning through their work. According to the research participants, the fundamental idea of becoming one through movement away from the dichotomy of disability and ability created great synergy in cooperative work. All agreed that perception of disability could never be a barrier to bringing about creative choreography or harmonizing among dancers.

### 4.2. Situation Model

The situation model is the stage in which the consequences thus far are summarized and integrated to promote the understanding of the core category and provide a more convincing explanation [41]. This study built the situation model for “creative intercorporeality in collaborative work of choreographers with and without disabilities” at the individual, artistic, and social levels. Figure 1 shows how the integrated approach that embraces intercorporeality is formed through collaboration between choreographers with and without disabilities. The oval-shaped lines in Figure 1 are marked as individual, arts community, and society & nation, and the dotted lines among the lines represent how the factors of each level affect the integrated approach that embraces diversity.

This situation model includes causal conditions as well as contextual, intervening conditions, action/interaction strategies, and consequences. Cases such as “new approach to the body”, “motivation for new idea”, “acquiring mutually integrated choreography”, “methodological agenda of support system”, that are hovering between the individual and artistic levels, that are hovering between the artistic and social levels indicate that the category appears in each one. Also, the consequences and stages of the process in which dancers participating in collaborative work of choreographers with and without disabilities accept creative intercorporeality and the core category were presented in order at the bottom of the figure.

## 5. Discussion

Until now, choreographers and performers who participate in disability art tended to improve their understanding of disability or gain new insights when they had more experience [21,42]. On the other hand, when the activity did not last, the dancers simply confirmed that they knew about the disability and had no opportunity to gain a new understanding [43,44]. This was supported by Marsh [45] who argued that there was not enough research evidence on adapted dance yet and that there was a lack of knowledge of collaboration among choreographers with and without disabilities. Ames, Benjamin, and Whatley have studied the vivid experiences of artists with disabilities and the dance activities of performers with disabilities [20,21,46], but they have shown that the development of adapted dance is uncertain unless all performers cooperate and show continued interest and participation [47,48]. This indicates that artistic opportunities and continuous activities in which the collaborations should continue [3,7,49].

This study was intended to present an academic discourse on the process of experiencing creative intercorporeality in the collaborative choreography process for choreographers with and without disabilities. This study analyzed the raw data on the experiences of ten dancers who participated in cooperative choreography between dancers with and without disabilities and derived a total of 190 conceptual words. Based on this, the axial coding method of Strauss and Corbin [41] was classified into 14 upper categories and 30 lower categories, and “creative intercorporeality in collaborative work of choreographers with and without disabilities” was derived as a core phenomenon forming the paradigm.

As shown in Figure 2, as factors influencing this core phenomenon, personal situations such as new motivation in choreography, collaboration of choreographers with and without disabilities and a new approach to the choreography of methodological agenda were found in common contexts. This led to questions such as “How far is the artistic possibility that human can express with their bodies?” and “Have we not limited the artistic beauty expressed with their bodies too much in stereotypes?” It was about the choreographer’s intellectual curiosity and value of beauty toward new and rare movements, as well as ventilation of socially prevalent prejudices. In particular, this study found that the beginning of cooperative work between choreographers with and without disabilities stems from creative attempts to escape stereotypes and challenges to themselves. It can be seen as an effort to expand the language of movement in various ways by not limiting the artistry of dance to the technical framework through “inclusive” dance of performers with and without disabilities [4,19].

However, unlike the motivation to participate, the process was never easy. “How do two choreographers with different thoughts and physical environments communicate with each other and solve problems of movement?” It said that the more they aimed for professional performance, the more difficulties they faced. In other words, constraint factors that mediate the synergy of collaborative choreography were derived. Cooperative choreography among choreographers with and without disabilities should avoid excessive technical expression in consideration of the physical risk of injury and resistance, difficulty and conflict of cooperation and obstruction and prejudice against disability that may arise. This dilemma in terms of working choreographers with and without disabilities has already been appeared in previous studies [4,20,21], so it can occur as much as possible and should be fully considered [24]. However, finding empirical insights on how to deal with the dilemma and defects and solving problems in the collaborative choreography process of the research participants are a meaningful work suggested by this study. 

This study focused on the action/interaction strategy that alleviated the interventional element. A “new approach” different from the general choreography process was found. Rather than entering the composition of the scenes and movements of the choreography related to the subject in earnest, each focused on exchanging existing activities in dance. They said they have studied the process of criticizing each other’s strengths and weaknesses about choreography and finding common denominators that can be matched by trying each other with their preferred styles for quite a long time. The process necessarily involved an emotional consensus of “mutual understanding and respect”. It can be seen that these mutual efforts presuppose not only the physical aspect but also emotional, cognitive, and social interactions in the cooperative choreography in the relationship between two choreographers.

The result of the current study can be presented as a situation model as “creative intercorporeality through collaborative work of choreographers with and without disabilities” at the individual, artistic, and social levels. In other words, through a series of processes called cooperative choreography, “creative intercorporeality” was concluded to have experienced meaningful achievements such as creative intercorporeality, discovery/rediscovery of potential, and sustainability of interaction.

This study attempted a qualitative approach focusing on the experience and interaction process of collaborative work among choreographers with and without disabilities in that the quality of the experience is important, emphasizing the essential role of oneself and other egos in social understanding. To explain this in detail, this study borrowed Shogo Tanaka’s concept of “intercorporeality” of “connection and empathy of cognition and behavior occurring in interrelationships between independent egos” [39].

In this study, the cooperative choreography between ones with and without disabilities had an important meaning in the serious creative process itself. In the cognitive process of trying to empathize with each other, the relationship between integration, empathy and trust was the basis, and this was implemented as an active and specific interactive behavior of unity through the body and harmonious choreography. In the process of cooperative choreography of choreographers with and without disabilities, “creative intercorporeality” could be defined as forming creative empathy and achievement through physical, emotional, cognitive, and behavioral interactions such as bond and intimacy between the two dancers.

This study was able to explain how choreographers with different conditions and situations in the context of diversity accept differences for the creative work of “choreography”, and how the behavioral synergy of the two individuals creates creative achievements. In particular, this study, based on the experiences of choreographers who participated in collaborative choreography toward the proposition of “possible dance”, is meaningful in that it practically showed the potential of inclusive dance and the possibility of inclusive gaze through creative dance.

## 6. Conclusions

There has been prejudice in the field of dance for a while. The participation of individuals with disabilities in dance was viewed as the domain of “community dance”, or explained as the mainstream’s “inclusive” attitude toward certain special people. But, it presupposes a contradiction that is already unilaterally limited, and diversity is not respected. In the challenge of implementing the art of ‘creative intercorporeality,’ there is not certain standards of how to move and express in dance. Moreover, disability is never a barrier in dance, an artist who delivers a message through various bodily movements, choreography, and a creative work develops the language of dance.

This study focused on the process in which cooperative work between choreographers with and without disabilities constantly forms emotional empathy and trust relationships through interactions for creative work. In particular, from the perspective of social cognitive theory, this process can be explained as presupposing ‘creative intercorporeality’ for the attitude of embracing each other’s perspectives in the interaction between the two dancers. Therefore, in the process of collaborative work of choreographers with and without disabilities, ‘creative intercorporeality’ can be defined as a creative attitude and perspective that expresses harmony, concurrency, consistency, and balance through physical, emotional, cognitive, and behavioral empathy between the choreographers.

The change in dancers’ perception of dance access to disability and methodology was significant in reaffirming the original aesthetic reflection of dance art as a creative expression of ‘choreography’ along with the trend of adapted dance. In particular, the intrinsic dynamism of creative intercorporeality in cooperative choreography reflects various trends as well as the philosophical background of dance. It is still prevalent to challenge blind cooperation work on the grounds of the movement’s own methodological approach or national support system without understanding or deliberation on ‘disability’. Disability art initiated against this background should be avoided. Along with this reflection, confirming the artistic performance and developing potential dancers with disabilities should also not be overlooked.

This study findings may make unique contributions to dance literature for the interactions between choreographers with and without disabilities, in that it provides insight to increase the value of dance arts. Along with the methodological attempts of various basis theories, it seems necessary to further discuss the a fundamental theory for the participation of individuals with disabilities in dance. Through this, it is suggested that a comprehensive review and discourse on dance for persons with disabilities should be presented, so that they keep participating in the art of dance.

## Figures and Tables

**Figure 1 ijerph-19-05548-f001:**
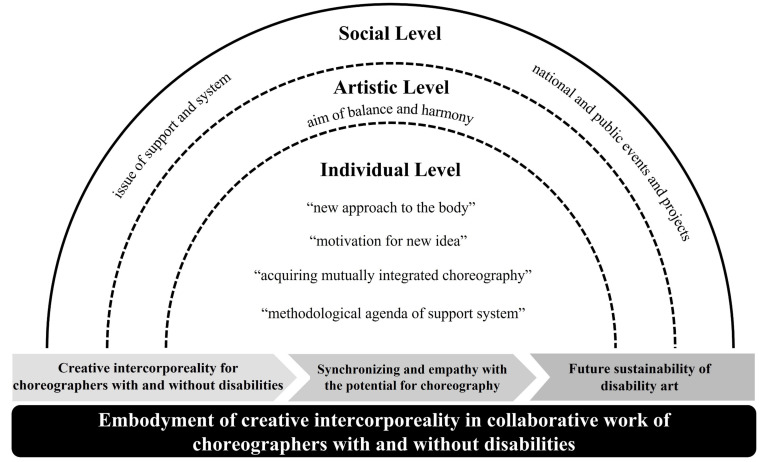
Situation model.

**Figure 2 ijerph-19-05548-f002:**
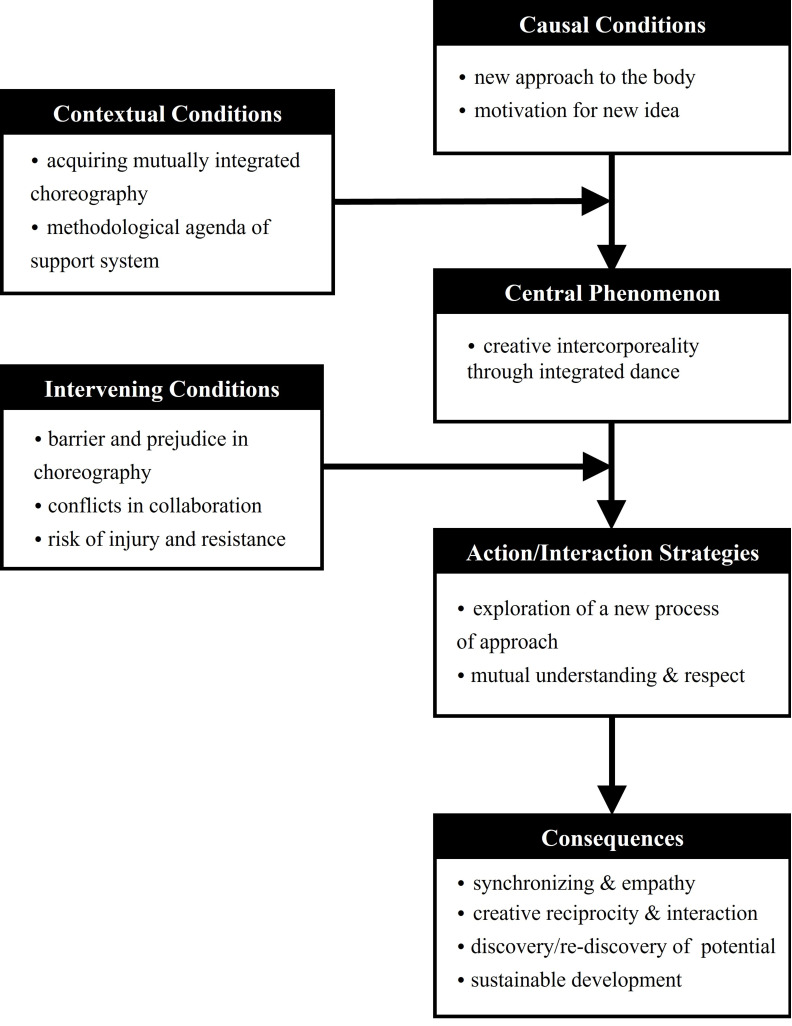
Paradigm model.

**Table 1 ijerph-19-05548-t001:** Research participants.

No.	Name	Gender	Age	Occupation	Career
A	Kim, S.	Female	30+	Dancer, Arts instructor	16 years
B	Shin, J.	Female	30+	Dancer, Arts instructor	15 years
C	Kim, A.	Female	40+	Dancer, Arts instructor	17 years
D	Han, J.	Female	40+	Leader of dance company	26 years
E	Lee, Y.	Male	40+	Professor	25 years
F	Song, J.	Male	40+	Professor	23 years
G	Kim, Y.	Male	40+	Leader of dance project	15 years
H	Kim, E.	Female	50+	Artistic director	31 years
I	Kim, H.	Female	50+	Leader of dance company	35 years
J	Jung, S.	Male	50+	Artistic director	36 years

## Data Availability

The data are not publicly available due to privacy issues.

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
