# Peer review of "Creative Intercorporeality in Collaborative Work of Choreographers with and without Disabilities: A Grounded Theory Approach"

_ijerph, 2022, doi:10.3390/ijerph19095548_

Round 1
Reviewer 1 Report
This paper offers a cogent report on a study of choreographic collaborations between able-bodied and disabled dancers in South Korea. It is well structured and offers a clear analysis using the grounded theory method, resulting in the authors' conclusion that greater awareness was obtained by both groups of artists when they collaborated. The article is well researched and offers a clear introduction, literature review, analysis, and conclusion with appropriate references. However, I would have liked to see a clear definition of what the authors mean by "disability art" and "creative intercorporeality"–terms that are central to their argument but are not clearly defined anywhere in the article. Also, the article could benefit from a thorough spell-check and a round of copy-editing for English language errors. Overall, it is an interesting study and one that will benefit the field of public health.
Author Response
Dear. reviewer
I appreciate the reviewer's detailed and detailed review opinion. Researchers tried to study the collaboration process between disabled and non-disabled dancers through this study at a time when the concept of "disability art" was changed and newly raised by each researcher. In the process, through the concept of "creative interpolation," attention was paid to how artists and experts with different characteristics and backgrounds can understand and cooperate with each other. The revised part of the received manuscript was marked in red.
An explanation of the concept presented by the reviewer was added, and the concept was revised and typos were reviewed through several revisions. I would like to express my deep gratitude to the reviewer who took a busy time to present his opinion for the improvement of our research.
If we give our research team a little more opportunity, we will do our best to correct more errors in the remaining time and refer to the results of several studies. Thank you very much.
Reviewer 2 Report
Strengths of the study are the appealing research question and its relevance for real life outcomes.
However, the sample is relatively small (10 individuals). Is the analysis representative and reliable? Is the sample size considered as study limitation?
The conceptual frameworks and how they will be advanced with the present study could be elaborated in more depth.
The relevance for individuals’ well-being and health could be illustrated in more detail.
Do the findings apply to other vulnerable groups, i.e. those with certain chronic diseases, frail older adults, etc.?
Author Response
Dear. reviewer
I appreciate the reviewer's detailed and detailed review opinion. Researchers tried to study the collaboration process between disabled and non-disabled dancers through this study at a time when the concept of "disability art" was changed and newly raised by each researcher. In the process, through the concept of "creative interpolation," attention was paid to how artists and experts with different characteristics and backgrounds can understand and cooperate with each other. The revised part of the received manuscript was marked in red.
The opinion that the number of samples asked by the reviewer is small is not insufficient because this study follows the paradigm of qualitative research and follows the technique of grounded theory. According to the grounded theory technique, each environment and context conditions were presented and the model was designed. As this journal emphasizes the importance of the environment and public health, I think it is a thesis topic and result that corresponds to this.
In order for the results of this study to be applied to other vulnerable groups with certin chronic causes, detailed research is necessary, and it can be found in common in that it is a study of unique characteristics, safety, and objects requiring attention.
If we give our research team a little more opportunity, we will do our best to correct more errors in the remaining time and refer to the results of several studies. Thank you very much.
Reviewer 3 Report
Into the abstract: It is important to reflect a final reflection that will enlighten the reader after this process of extraction of categories and subcategories.
I have enjoyed the article very much and it has been very inspiring for me in the theoretical foundation, in the introduction.
But everything has begun to get complicated from the methodological part.
I have missed knowing what specific questions were asked in the 4 areas of comprehension. In other words, I still need to understand what process of concretion was followed in the questions that were sent
I do not understand those epigraphs of the results that they want to indicate.
In relation to the results and discussion part, I find the size of the results a little disproportionate in relation to the discussion with only 6 bibliographic references of the 50 that are in the article, the other 44 are between the introduction (42) and 2 in methodology.
There is a lack of a more concretizing final conclusion of all the results and concretion
Please , I ask the authors to read the minimum necessary that references are requested and read again the regulations of the magazine
Author Response
Dear. reviewer
I appreciate the reviewer's detailed and detailed review opinion. Researchers tried to study the collaboration process between disabled and non-disabled dancers through this study at a time when the concept of "disability art" was changed and newly raised by each researcher. In the process, through the concept of "creative interpolation," attention was paid to how artists and experts with different characteristics and backgrounds can understand and cooperate with each other. The revised part of the received manuscript was marked in red.
From the review opinion pointed out by the reviewer, it was found that additional explanations of the research results, reviews of more references, and conclusions were needed. Accordingly, the research team considered and revised each part of the study. The revised part of the received manuscript was marked in red.
If we give our research team a little more opportunity, we will do our best to correct more errors in the remaining time and refer to the results of several studies. Thank you very much.